# Organobentonite Binder for Binding Sand Grains in Foundry Moulding Sands

**DOI:** 10.3390/ma16041585

**Published:** 2023-02-14

**Authors:** Beata Grabowska, Sylwia Cukrowicz, Artur Bobrowski, Dariusz Drożyński, Sylwia Żymankowska-Kumon, Karolina Kaczmarska, Bożena Tyliszczak, Alena Pribulová

**Affiliations:** 1Faculty of Foundry Engineering, AGH—University of Science and Technology, Reymonta 23, 30-059 Krakow, Poland; 2Department of Materials Engineering, Faculty of Materials Engineering and Physics, Cracow University of Technology, 37 Jana Pawła II Av., 31-864 Krakow, Poland; 3Faculty of Materials, Metallurgy and Recycling, Technical University of Košice, Letná 9, 042 00 Košice, Slovakia

**Keywords:** montmorillonite, organobentonite, poly(acrylic acid), shungite, lustrous carbon, foundry engineering

## Abstract

A series of studies related to the production of organobentonite, i.e., bentonite-poly(acrylic acid), and its use as a matrix grain-binding material in casting moulding sand is presented. In addition, a new carbon additive in the form of shungite was introduced into the composition of the moulding sand. Selected technological and strength properties of green sand bond with the obtained organobentonite with the addition of shungite as a new lustrous carbon carrier ($R_{c}^{w}$, $R_{m}^{w}$, $P^{w}$, $P^{w}$, $P_{D}$) were determined. The introduction of shungite as a replacement for coal dust in the hydrocarbon resin system demonstrated the achievement of an optimum moulding sand composition for practical use in casting technology. Using chromatographic techniques (Py-GC/MS, GC), the positive effect of shungite on the quantity and quality of the gaseous products generated from the moulding sand during the thermal destruction of its components was noted, thus confirming the reduced environmental footprint of the new carbon additive compared to the commonly used lustrous carbon carriers. The test casting obtained in the mould of the organobentonite moulding sand and the shungite/hydrocarbon resin mixture showed a significantly better accuracy of the stepped model shape reproduction and surface smoothness compared to the casting obtained with the model moulding sand.

## 1. Introduction

Bentonite is the primary and most widely used inorganic matrix grain binding material in foundry moulding compounds worldwide. The reason why bentonite is used in various industries, including foundry applications, is not only because of the presence of deposits of this material on almost every continent but also because of its excellent adsorption and absorption properties, which result from its crystallographic structure [1]. The main constituent of bentonite is montmorillonite (MMT) with an example formula: Na^+^_0.67_[Al_3.33_Mg_0.67_]^−^_(0.67)_Si_8_O_20_(OH)_4_·*n*H_2_O which crystallises in a single-stranded arrangement as compact, finely scaly clusters. It belongs to the group of clay minerals of phyllosilicates with triple-layer packets of 2:1 type of structure, in which the octahedral layer is located between two tetrahedral layers facing each other. The oxygen anions (O^2−^) on the tetrahedral vertices face inwards and surround, together with the hydroxyl groups, the aluminium (Al^3+^), magnesium (Mg^2+),^ and iron (Fe^2+^) cations, forming the octahedral layer. The two tetrahedral layers are linked by strong ion-atom bonds to one octahedral layer, forming a structural unit. The thickness of the montmorillonite layer is about 1 nm and the length is in the range 100–150 nm. Cations can penetrate between the packets, as well as water molecules [2]. Thus, MMT is capable of surface adsorption as well as absorption of water molecules, manifesting itself as swelling [3]. Depending on the type of predominant exchangeable cations, a distinction is made between calcium (MMT-Ca), sodium (MMT-Na), potassium (MMT-K), and magnesium (MMT-Mg) montmorillonites, among others. Analogous to montmorillonite, it has been accepted to call bentonites: calcium bentonite (containing mainly MMT-Ca in its composition), sodium bentonite (containing mainly MMT-Na in its composition), potassium bentonite (containing mainly MMT-K in its composition) and magnesium bentonite (containing mainly MMT-Mg in its composition).

About 70% of cast iron castings are made in bentonite-bonded moulding sands in Poland and abroad. Commercial bentonites for the foundry industry and those available on the domestic market include Bentonite S (S&B Industrial Minerals, Oberhausen, Germany), Geko Optimum (Süd-Chemie, Munich, Germany) or bentonite Specjal (ZGM “Zębiec” S.A, Starachowice, Poland). In order to obtain a full-quality casting without defects, a carbon or organic additive (lustrous carbon carrier) is often introduced into moulding sands with bentonite. This additive improves the technological properties of the moulding sand, including enhancing the binding force of the matrix grains and at the same time reducing its oscillation. It also improves the surface quality of castings. The carbon-bearing material is, in the classical approach, coal dust, but also mixtures of organic substances (e.g., hydrocarbon resins) and graphite are used [4,5,6]. The amount of bentonite introduced into the moulding sand depends on its properties and is usually between 5–10% bentonite. Apart from bentonite, the main components of the sand are grain matrix (85–95%), water (2–5%), and carbon-bearing material (3–8%). The lustrous carbon carriers present in the moulding sand composition undergo thermal degradation with limited access to oxygen when the mould is poured with liquid metal. The thermal decomposition results in the formation of pyrolytic carbon, which is a mixture of amorphous and so-called ‘lustrous carbon’ [4,7].

However, the introduction of carbon-containing materials into green sands with bentonite generates a new environmental problem. The presence of carbon carriers causes increased emissions of harmful substances, including aromatic hydrocarbons, during the pouring of the mould with a liquid metal alloy. Hence, research is being conducted into the development of bentonite-bonded moulding sand technology with new or modified carbon carriers to reduce their negative impact on the environment [8]. However, it should be borne in mind that the mechanism of thermal decomposition of the carbon additive depends on its type, structure, and quantity. The resulting gas phase contains a mixture of hydrocarbons and has a reducing composition. Under these anaerobic and reductive conditions, a fraction of carbon, so-called ‘lustrous carbon’, is formed, followed by its physical adsorption onto the surface of quartz matrix grains, also heated to temperatures in the range 650–1200 °C. The formed layer of ‘lustrous carbon’, about 0.1 μm thick, adhering to the matrix of sand grains, constitutes a barrier against the penetration of liquid metal between the matrix grains, as it is non-wettable by liquid metal, prevents its mechanical penetration into the moulding sand and chemically isolates the mould material from the metal. In this way, it prevents chemical reactions between metal and non-metal oxides contained in the alloy (liquid metal) and the moulding sand components, thus ensuring that the surface of the casting is smooth and free of burns from the sand [9].

The carbon additives currently used in bentonite-bonded sand technology have a theoretical ability to form lustrous carbon in a fairly wide range of 7–60%, including coal dust up to 11%, hydrocarbon resins that are replacements for coal dust up to about 50–60%, but as practice has shown, the optimum level is in the range of 15–20%. The use of substitutes as precursors for the formation of ‘lustrous carbon’ allows a significant reduction of these additives in the moulding sand composition, while obtaining good quality castings, without defects and especially with good surface quality. In comparison, coal dust is added to the moulding sand at around 4–5% and the replacement at 1.0–1.5%. In addition, the reduction in the amount of carbon additive in the moulding sand results in a reduction in the emission of harmful gaseous products from their destruction [10].

However, the use of lustrous carbon carriers is not a guarantee of obtaining good casting surface quality in an environmentally friendly technological process. The use of carbon dust or substitute in the case of their excessive amount in the moulding sand may cause the formation of casting defects such as ‘breaks in metal continuity’, or defects of gaseous origin in castings, such as ‘blisters’, ‘punctures’. In addition, the currently used carbon additives, mainly hydrocarbon resins, are characterised by a fairly high level of emission of harmful substances into the atmosphere (min. aromatic hydrocarbons from the BTEX group: benzene, toluene, ethylbenzene, xylenes) [11,12].

The most common bentonite moulding compound additives in Poland are ready-made mixtures of coal dust and other lustrous carbon carriers (e.g., commercial Kormix, ekosil) or bentonite-Kormix mixtures, where the proportion of carbon additives is from 75% upwards. The emission of harmful substances from such a mix depends on the additive used in moulding sands, its composition and origin, and the temperature used [8]. Compound emissions from such a moulding sand include mainly aromatic hydrocarbons, carbon monoxide and dioxide, and methane, which is not good for the environment.

The earliest patent application US 3,666,706 from 1972 contains the first attempts to substitute coal dust with other compounds to improve the surface quality of castings. This is an additive to the moulding compound in the form of a non-foamed and non-substituted hydrocarbon polymer in a finely ground form with a particle size of less than 0.3 mm in an amount of 0.5% to 3%, to provide a lustrous carbon-forming component instead of coal dust in the moulding compound composition with 6–8% bentonite as a binder. Polystyrene was used here, which releases a complex mixture of polycyclic aromatic hydrocarbons (PAHs) when burned at 800–900 °C, hence it is harmful to the environment [13].

An analysis of the state of the art shows that many types of carbon additives are known to be introduced into bentonite-bonded moulding sands [7,14,15,16,17,18,19,20]. However, these moulding sands do not guarantee a full-quality casting with environmentally friendly technology. The search for effective and more environmentally friendly substitutes for them has not yet yielded fully satisfactory results. However, taking advantage of the extensive knowledge in the area of properties and applications of various types of composites with bentonite (so-called organoclay), a possible solution for the research team was the structural modification of montmorillonite [21,22].

The specific crystalline structure and the highly chemically reactive surface of layered MMT allow its properties to be altered by reacting with selected organic compounds. These reactions can take place through electrostatic forces via ion exchange, the formation of secondary bonds leading to the adsorption of neutral molecules or covalent bonds between the reactive surface groups of the mineral and the organic molecules of the modifier as a result of the grafting process [23]. Regardless of the mode of interaction, the formation of MMT structures intercalated with macromolecular compounds offers the opportunity to develop a new precursor with the desired structure of shiny carbon as a result of thermodestruction of the organic part of the composite without losing the key binding properties of the casting binder. The proper selection of the organic modifier is a key step in the mineral modification process determining the level of environmental safety of the new inorganic-organic binder, which would be capable of binding mineral matrix grains in synthetic moulding sand technology while acting as a substitute for common carbon-forming additives. Therefore, poly(acrylic acid) (PAA) was chosen because of its promising physicochemical characteristics challenging the possibility of generating harmful compounds during high-temperature pyrolysis. PAA has reactive functional groups in the form of carboxyl groups, which provide the macromolecule with a high negative charge density upon dissociation. This makes them, together with poly(sodium acrylate), one of the most commonly used water-soluble anionic polyelectrolytes, e.g., in the production of hydrogels, so-called superabsorbents, and ion exchange resins, or as dispersing and binding agents [24,25,26]. Therefore, possessing properties valuable from the point of view of the requirements for casting binders in synthetic moulding sand technology, the selected polymers represent a promising material for the modification of the structure of the main component of bentonite [27]. However, the amount of carbon in the resulting organobentonite does not guarantee that casting with an adequate surface quality will be obtained, so a new carrier of lustrous carbon in the form of shungite was introduced into the moulding sand. At the same time, it is a replacement for those already used in casting. Shungite is an amorphous substance, the main component of which is non-crystalline carbon (up to 99%) in the form of multi-layered hollow globules ≤ 10 nm in size. According to one definition, shungite is a type of amorphous carbon classified as anthracite and graphite [28], although researchers are still not in agreement on this, especially as shungites show different characteristics depending on the deposit [29]. Based on observations of graphitic layers in shungite, shungite carbon was found to have a structure similar to fullerene [30]. The data also show that shungite globules present aggregates of six- and five-layer graphene-like stacks [31]. Based on the proportion of carbon, it is classified into five groups: shungite-1 (98–100 wt.% C), shungite-2 (35–80 wt.% C), shungite-3 (20–35 wt.% C), shungite-4 (10–20 wt.% C) and shungite-5 (<10% wt.% C), with no clear distinction between pure shungite and shungite rocks (e.g., shungite dispersed in rock deposits is also included in the classification). In addition to carbon, shungite also contains microcrystalline (<1 to 10 µm) mica, quartz, albite, and other minerals. The chemical composition of shungite is determined by the location of the deposit. Hence, shungite may contain SiO_2_, Al_2_O_3_, TiO_2_, Fe_2_O_3_, MgO, K_2_O and S [32]. Due to its specific composition, shungite is considered a natural composite material with potential sorption properties. Studies confirm that shungite is capable of sorbing many organic compounds and iodine. Shungite (pure) is a non-graphitised carbon that is very resistant to heat treatment. Significant changes within the structure occur at temperatures above 2100 °C, which is probably related to an increase in coherent dispersion regions and a reduction in the distortion of the carbon layer. On the other hand, shungites in rock deposits are characterised by such fragmentation and dispersion of minerals within the structure and a large contact area between carbon and mineral constituents that provide high carbon and silicon content, which determines the possibility of their mutual crystallogenesis, including the formation of carbides and silicides in the temperature range from 1400 to 1600 °C [33].

Currently, the area of application for shungite, taking into account the characteristics of this material, includes metallurgy (where it is introduced into the process instead of coke and graphite) and the production of non-stick coatings, radar structures, heat-insulating and high-resistance materials, catalysts for various chemical processes, and fillers for acid-resistant and refractory materials [32,33].

This article presents a series of studies starting from the identification of the methodology for obtaining organobentonite, through the study of selected strength and technological properties of classic moulding compounds bond with the produced organobentonite with the addition of a new carbon carrier in the form of shungite. Finally, an ecological assessment of the casting production process using the technology developed by the research team was carried out, together with a qualitative assessment of a trial casting.

## 2. Materials and Methods

### 2.1. Basic Materials

Sigma-Aldrich (St. Louis, MO, USA) poly(acrylic acid) was used to produce organobentonite (Table 1).

Two forms of mineral fraction in the form of calcium bentonite with the trade name SN and sodium bentonite with the trade name Specjal (S) (ZGM “Zębiec” S.A., Starachowice, Poland) were used in the course of the research. SN bentonite was the starting mineral for obtaining the established organobentonites. S bentonite is popular and widely used in synthetic moulding sand technology (green sand) as an inorganic binding material. Additionally, SN and S bentonites served as reference materials for the produced organobentonite. The grain diameter of bentonite in both cases was less than 0.056 mm. The chemical composition and basic physicochemical properties of SN and S bentonites are summarised in Table 2. More details and information on physicochemical of the composition of S and SN are provided in the publication [27].

Table 3 lists the initial components used to prepare the green sand, which was then sent for further testing. The table also includes a model moulding sand in the form of a bentonite/Kormix mix (hereafter referred to as S/Kormix) for practical use.

### 2.2. Preparation of Organobentonites

Previously prepared PAA solutions with percentage concentrations of 20% were introduced into mineral suspensions (dispersed in a laboratory stirrer; 300 rpm, 3 h) at a ratio of 5 g of SN bentonite per 100 mL of distilled water.

Aqueous polymer solutions were prepared by dissolving the appropriate amount of PAA (5% by weight of mineral, respectively) in 20 mL of distilled water. The mixtures were homogenised for 6 h in a laboratory stirrer at 300 rpm and then left for a modification time of one week.

The mixing operation was repeated and the resulting dispersions were centrifuged in a laboratory centrifuge (8000 rpm, 12 min). After separation from the unreacted polymer, each modifier precipitate was dried to constant weight at 105 °C and then ground in an agate mortar. A first series of organobentonites were obtained with the acronyms: SN/5PAA, SN/15PAA, SN/25PAA.

### 2.3. Preparation of Organobentonite-Bonded Moulding Sand

At the stage of preparing SN/PAA organobentonite-bonded moulding sands (organobentonite-bonded green sands), it was found that a higher proportion of the polymer adversely affected the moulding sand already at the stage of mixing. The high viscosity of the system in contact with water hindered the accurate homogenisation of all the components, which resulted in significant deterioration of the technological and strength properties. Therefore, moulding sands were prepared with organobentonite of the lowest polymer content in the system (i.e., SN/5PAA). In this case, the process of mixing the components occurred effectively. The model moulding sand was bonded with unmodified calcium bentonite (SN).

Due to the risk of insufficient efficiency of the polymer in organobentonite as a carbon additive in the technology of synthetic moulding sand, technological and mechanical tests of SN/5PAA bonded moulding sand with a mixture of popular lustrous carbon carriers: coal dust and synthetic resin were also carried out. In addition, shungite was used as a new carbon additive in the field of foundry engineering, whose effectiveness and reduced environmental footprint were confirmed in the patent application (invention project entitled: “Bentonite-bound moulding sand with carbon additive”, application no: P.439688). Two moulding compounds were prepared: a base compound bond with SN/5PAA modifier and a reference compound bond with SN bentonite.

The moulding compound was prepared by mixing the ingredients in a WADAP LM-1 type rotary mixer: 100 parts by weight of silica sand, six parts by weight of binder material and water. Detailed data on the proportions of the individual components are contained in a collective table providing information on the composition of all the moulding sands analysed in this work (Table 3). The technological and mechanical properties of the moulding sand were assessed depending on the water content, so its share in the moulding sand was not taken into account in Table 4. The composition of the moulding sand was determined on the basis of literature data [4,34]. Mixing of matrix and free-flowing binder material was carried out for 1 min, then after the introduction of water, the whole mixture was mixed for another 3 min. The moulding sand was sieved through a 4 × 4 mm mesh sieve. After forming standard shapes, a series of determinations of selected technological and mechanical properties were made. Due to the research on changes in the analysed indices of the moulding sands depending on their humidity (by adding water), the moulding sand was put back into the mixer bowl, another portion of water was added and the mixing process was repeated for 3 min. The tests were carried out at air humidity of about 27% and ambient temperature in the range of 24–26 °C.

Two series of moulding sand with carbon additives were prepared. The first, in which SN/5PAA acted as the binding material, was prepared with mixtures of coal dust and hydrocarbon resin (P/HCR), shungite and hydrocarbon resin (Sz/HCR), and coal dust and shungite (P/Sz). The second batch of SN-bond moulding sand constituted the reference moulding sand to the first batch of sand and was prepared with the addition of analogous coal mixtures.

The methodology for preparing the moulding sand was analogous to that described above, except that an appropriate amount of a given mixture of carbon additives was initially introduced into the rotary mixer along with the matrix and binder material. The amount of shungite was determined by the carbon content compared to the coal dust and HCR resin, hence its mass proportion in the moulding sands was proportionally higher than that of the other glossy carbon carriers. Table 4 summarises the compositions of the prepared moulding sands.

Table 4 also includes the composition of the moulding sand, in which the role of the binding material was played by commercial sodium bentonite and the carbon carrier was a bentonite-Kormix mixture. S and S/Kormix bentonite-bonded moulding sands were prepared similarly to moulding sands containing unmodified SN bentonite with carbon carriers and were a real standard of technological and strength properties that should characterise a high-quality material binding grains of mineral matrix in the technology of synthetic moulding sands.

### 2.4. Preparation of Laboratory Samples

Standardized cylindrical shapes were made from the prepared moulding sand to determine their green compressive strength ($R_{c}^{w}$) and green tensile strength ($R_{m}^{w}$). The same type of laboratory sample was used to measure the permeability ($P^{w}$), friability ($S^{w}$) and flowability ($P_{D}$) of green sands containing calcium bentonite and potential binding material in the form of obtained organobentonite. Three cylindrical samples (⌀50 × 50 mm) were used for each determination particular property. All the samples were formed manually in a non-split cylindrical matrix which, together with the weighted green sand portion, was placed under the foot of the LU-type laboratory rammer. Compaction was carried out by hitting the rammer three times with a 6.66 kg rammer weight from a height of 50 mm. Before each batch of shaped pieces was made, the amount of green sand needed to obtain a standard cylindrical shaped piece with a height within the specified limits of the nominal height of the shaped piece was adjusted. The cylindrical shapes were then removed from the matrix using an ejector.

### 2.5. Investigations of Green Sands

#### 2.5.1. Investigations of Selected Technological and Mechanical Properties

Mechanical tests of moulding sand were carried out using a universal apparatus for determining mechanical properties, type LRu-2e made by Multiserw Morek (Marcyporęba, Poland). The green tensile strength $R_{m}^{w}$ and green compressive strength $R_{c}^{w}$ of the moulding sand were measured by appropriate clamping of jaws and holders. The determination of $R_{c}^{w}$ was performed by placing the shaper vertically between the compression jaws with a diameter corresponding to the diameter of the specimen base. An axial pressure was applied to the sample until it was broken. The determination of $R_{m}^{w}\;$was carried out by destroying the sample under tensile force. The measuring range was 0–22.4 N/cm^2^ for $R_{m}^{w}$ and 0–130 N/cm^2^ for $R_{c}^{w}$. The strength values were expressed in the SI unit: MPa. The tests were carried out according to the PN-83/H-11073/EN standard.

Permeability in a non-dried state ($P^{w}$) was performed by the fast method on electrical apparatus type LPiR1. The permeability values were expressed in the SI unit: m^2^/Pa·s. Permeability was determined for standard cylindrical samples according to the PN–80/H–11072 standard.

The friability ($S^{w}$) tests were carried out on an LS (Huta Stalowa Wola production) apparatus and consisted in determining the relative mass loss of a cylindrical sample after rolling it over a pair of ⌀50 rollers (60 rpm for 5 min) while being heated by an infrared lamp. The determination was carried out in accordance with the BN-77/4024–02 standards.

Flowability ($P_{D}$) of the moulding sand was carried out according to the method of H.W. Dietert and F. Valtier using a hand rammer with a fixed sensor for the degree of deformation of a standardised cylindrical-shaped samples between the fourth and fifth blow of the weight of a standard hand rammer. It should be noted, that none of the methods for testing the flowability of green sands have been standardised globally to date.

The numerical value of Dietert’s flowability, expressed in %, was determined from the Equation (1):(1)$$P_{D}=100-40x$$
where: *x*—the loss of height of the cylinder, mm.

The determination of compactability (*Z*) was based on the measurement of the percentage reduction in the column height of the mass loosely poured into the metal tube under the influence of constant pressing pressure.

#### 2.5.2. Pyrolysis Gas Chromatography-Mass Spectrometry

The identification of gaseous products formed during the thermal degradation of shungite was carried out using the pyrolysis gas chromatography-mass spectrometry method (Py-GC/MS).

In this method, the following instrumentation was used: the pyrolyzer “Py” Pyroprobe 5000 (CDS Analytical Inc., Oxford, PA, USA), the gas chromatograph “GC” Focus GC (Thermo Scientific, Waltham, MA, USA), coupled with the mass spectrometer “MS” Focus ISQ (Thermo Scientific). The study is based on transforming a solid sample (2–3 mg) into a gas (also the so-called ‘fast pyrolysis’) by heating an atmosphere of inert gas (helium) in a pyrolyzer, which is accompanied by thermal decomposition. The obtained mixture of compounds (pyrolysate) is separated on a chromatographic column in a chromatograph. The gas chromatography “GC” conditions were as follow: an initial temperature of 40 °C (3 min hold) was raised at 3 °C/min to 100 °C (3 min hold) and then at 20 °C/min to 250 °C (3 min hold) using constant helium flow of 1 cm^3^/min during the whole analysis. The temperature of the split injector was 250 °C and the split ratio was 1:30. The transfer line temperature was 250 °C. The EI ion source temperature was kept at 250 °C. The ionization occurred with a kinetic energy of the impacting electrons of 70 eV. The gaseous products were identified based on the mass spectral library NIST MS Search 2.0 Libera (Chemm. SW, Version 2.0, Fairfield, CA, USA) using the Xcalibur program (ver. 2.2).

#### 2.5.3. Determination of the Gasses Emission Released from Moulding Sands

Tests of gasses emission of moulding sands prepared with the use of organobentonite and mixtures of carbon additives selected on the basis of the results of research on technological and mechanical tests were carried out in accordance with the procedure developed at the Faculty of Foundry Engineering AGH [35]. For this purpose, standard cylindrical shapes were prepared according to the method described above, with the difference that the moulding sand compaction was carried out by hitting the rammer weight five times. The shapes were dried to constant weight at a temperature of 100 °C, and then stored in a desiccator until it is properly measured. Before placing the shaped piece in the measuring system, it was weighed with an accuracy of 0.01 g. The pouring temperature with cast iron was 1350 °C. The volume of gasses evolved during the process of pouring the shapes with liquid metal and was recorded until the emission from the measuring sample ceased. The results of the analysis were averaged from three measurements.

#### 2.5.4. Gas Chromatography

Determination of gaseous products (hydrocarbons from the BTEX group) released from the tested moulding sands was carried out using gas chromatography. The test methodology was consistent with the described patent [35] and the literature [8,36,37]: gasses were released during pouring the measuring shapes with liquid metal adsorbed on a layer of activated carbon. The adsorbed products were eluted with diethyl ether ((C_2_H_5_)_2_O) obtaining research material for the quantitative analysis of aromatic hydrocarbons. The amount of solvent needed to fully extract the compounds from the BTEX group of the gasses adsorbed on the column with activated carbon was 40 mL (in four cycles of 10 mL). The extracts obtained for individual samples were analysed by gas chromatography with a flame ionization detector FID according to the following process parameters: an initial temperature of 40 °C (3 min hold) was raised at 10 °C/min to 150 °C (5 min hold) using constant helium flow of 1 cm^3^/min during the whole analysis. The split ratio was 1:30 and the sample size was 1 µL. The separated gaseous products were identified using the Thermo Scientific TRACE Ultra gas chromatograph, equipped with a 30 m long RTX 5MS (ResteK, Bellefonte, PA, USA) chromatographic column with an internal diameter of 0.25 mm. The internal standard was a mixture of BTEX dissolved in diethyl ether prepared in the appropriate weight ratio.

#### 2.5.5. Making of Test Casting

A four-stage stepped model was used to make the test casting of cast iron with the “stepped test casting” method with the thickness of the steps: 3, 5, 10, 20 mm in the horizontal position (Figure 1). Foundry moulds were made of moulding sands:with the participation of organobentonite SN/5PAA, selected on the basis of the test results of the first stages of work andwith a bentonite-Kormix mixture as reference moulding sand.

The experimental smelting was carried out in an induction furnace. The pouring temperature of cast iron was 1350 °C. The obtained cast iron had the following chemical composition (Table 5):

The stepped casting model was moulded horizontally in order to reduce the influence of metallostatic pressure on the test result, i.e., the surface quality of the casting.

## 3. Results

### 3.1. Tests Results and Analysis of Technological and Strength Properties

The results of tests on green moulding sand with SN/5PAA organobentonite were compared with the properties of green sands made with unmodified calcium bentonite SN and sodium bentonite S.

A deterioration in the strength properties of SN/5PAA moulding sand was noted in comparison with SN and S moulding sands (Figure 2). The organic modification of bentonite reduced the compressive strength values moulding sand reaching the maximum value of 0.068 MPa with the highest value of $R_{c}^{w}$ equal to 0.098 MPa for SN moulding sand and 0.140 MPa for S moulding sand in the moisture content range of 1.6–1.7%. Thus, it appears that the contribution of the polymer of acrylic polymer does not influence the improvement of binding properties of silica matrix grains as expected in the system with MMT. The polymeric envelope around the bentonite grains, confirmed by BET studies of organobentonites [38], most likely inhibits the absorption of water by the bentonite grains, limiting the binding properties to the sorption capacity of the polymer only.

Figure 3 shows the results of determining the permeability and compactability of the analysed green sands. The best permeability was found for the green sand in which the binder was organic modified bentonite (SN/5PAA). This is a favourable effect because the low permeability of the moulding sand can be the cause of casting defects.

There was a shift in the compactability index towards a higher moisture content of the moulding sands with SN/5PAA compared to SN and S moulding sands, which may be due to the presence of a fine polymer fraction increasing the viscosity of the system as the proportion of water in the moulding sand increases. However, this slight shift is not significant (a difference of 0.2% in moisture content). It is also within the recommended value of compactability (*Z*) for green sands (35–45%) and fully correlates with the maximum strength obtained in this moisture range. Therefore, it should not have a negative impact on the technological process of mould preparation.

Figure 4 shows the results of Dietert’s flowability and the friability of moulding sands bonded with SN/5PAA, SN, and S materials.

The lowest Dietert flowability, in this case characterising the ability of grains to move under the influence of external compaction forces, is shown by organobentonite-bonded green sand. The $P_{D}$ value at the moisture content at which the moulding sand shows the best strength properties (about 2.0–2.2%) is comparable to the flowability of the green sand bond with sodium bentonite and is within the statistical error limit.

The determination of the friability showed a similar behaviour to all tested systems at lower moisture contents. In the range of “working humidity” (maximum green strength and permeability), the green sands with organobentonite SN/5PAA shows lower friability, which indicates the preservation of favourable technological properties of the moulding sands at the limit of contact between the mould and liquid metal.

### 3.2. Organobentonite-Bonded Moulding Compounds with Carbon Additives

Figure 5, Figure 6 and Figure 7 are a summary of strength and technological properties as a function of moisture content of the moulding sands bond with unmodified and poly(acrylic acid) modified bentonite, taking into account respective mixtures of carbon additives, i.e., coal dust (P), resin (HCR) and shungite (Sz) with compositions corresponding to those given in Table 4.

The role of the model binder material was fulfilled by the bentonite-Kormix mixture.

The relationship between the compressive and tensile strengths of the green sands is shown in Figure 5.

A favourable influence of the presence of shungite, especially as a substitute for coal dust, on the green strength properties of moulding sands was recorded. This fact is visible both for green sand bonded with unmodified and organically modified bentonite. Among the moulding sands in which SN played the role of the binding material, the best green compressive strength is characterised by SN/Sz/HCR with a maximum of 0.125 MPa at a moisture content of 1.8%. The highest $R_{c}^{w}$ value was recorded for SN/5PAA/Sz/HCR reaching 0.13 MPa at 1.8% moisture content. The same green sand (SN/5PAA/Sz/HCR) also has the best green tensile strength, except that the maximum $R_{m}^{w}$ is shifted towards higher moisture compared to the other mixtures and is approximately 0.022 MPa. Such good green tensile properties of SN/5PAA/Sz/HCR are most likely due to the fact of obtaining a system in which there is the greatest compactability of the moulding sand components, especially SN/PAA organobentonite with shungite. The structure and physicochemical properties of shungite positively influence the strength of interaction with the modifier and at the same time with the matrix grains, which is manifested by an increase in the $R_{c}^{w}$ and $R_{m}^{w}$ indices of the tested green sands.

Figure 6 shows the results of the permeability and compactability determinations for the analysed green sands.

Similarly to the strength properties, a favourable effect of the presence of shungite on the permeability of the green sands with carbon additives was recorded compared to the moulding sands with a mixture of coal dust and HCR resin. Among the moulding sands, the highest permeability value was recorded for SN/P/Sz (295 × 10^−8^ m^2^/Pa·s) and SN/5PAA/Sz/HCR (285 × 10^−8^ m^2^/Pa·s) at a moisture content in the range 2.2–2.4%, at which the highest tensile strength value was also obtained.

A shift in the compactability index of SN and SN/5PAA green sand with carbon additives towards higher moisture contents is noticeable. However, the density of moulding sand with organobentonite and shungite/HCR resin mixtures reach lower values compared to the rest of the tested systems, but only up to a moisture content of 2.2%. As the moisture content increases, the compactability increases, obtaining in the range of the highest strength and permeability (2.2–2.4%) better values than for the other moulding sands, e.g., SN/P/HCR, falling within the recommended *Z*-value limits for synthetic moulding sands.

On the basis of liquidity determinations of moulding sands of a given composition, it was found that the moulding sands in which calcium bentonite was the binding materialshowed better flowability than those with organically modified bentonite (Figure 7).

The difference in the flowability of the moulding sands is due to the physico-chemical properties of the binding materials used. Poly(acrylic acid) partially dissolved in water takes the form of a viscous solution, which together with bentonite forms agglomerates surrounding grains of silica sand. As a consequence, SN/5PAA moulding compounds are less flowable. Moulding sands bonded with SN and SN/5PAA and with respective mixtures of carbon additives are characterised by a similar level of flowability. At the same time. It should be noted, however, that moulding compounds containing shungite and HCR resin (SN/Sz/HCR and SN/5PAA/Sz/HCR) show the lowest tendency to settle at higher moisture contents (above 2.5%) than all the other tested systems.

### 3.3. Qualitative Analysis of Thermal Decomposition Products: Py-GC/MS Method

Figure 8 is a list of compounds identified in the pyrolysis process of two popular carbon additives introduced to the moulding sand: coal dust (Figure 8a) and HCR hydrocarbon resin (Figure 8b). The temperature of the pyrolysis process of 1100 °C corresponded to the temperature of pouring the casting mould with liquid cast iron. For the better quality graphs, only two operating value temperatures, 250 °C and 1100 °C, were taken into account. The chromatograms present a relation between time retention R_T_ and the intensity of tested samples. For HCR resin, chromatograms have been combined taking into account the four operating values of the process temperature: 250 °C, 500 °C, 700 °C and 1100 °C to present the growing diversity and quantities emitted during the gas pyrolysis process. The products of pyrolysis of coal dust distinguished two compounds at 250 °C and three main compounds at 1100 °C. Regardless of the process temperature first carbon dioxide (CO_2_) was released at the RT retention time: 2.17 and 2.24 min. Considering the intensity of peaks in a much smaller amount, toluene appeared at ~7.80 min for 250 °C and 1100 °C. At a higher temperature, additional low-intensity peaks are noticeable, of which the clearest was a peak at R_T_ equal to 4.79 min indicating the presence of benzene in gas coal dust thermal degradation products.

The HCR resin pyrolysis process generated from four at 250 °C to twelve at 1100 °C of volatile compounds. As the temperature increases, the growing variety of gasses and their participation in all emitted substances are visible. There was no significant release of toxic gasses of aromatic compounds in the temperature range of 250–700 °C. The highest emission of compounds expressed in the intensity of peaks corresponding to the presence of decomposition material of cyclical unsaturated hydrocarbons, such as cyclopentadien and aromatic hydrocarbons in the form of benzene, toluene, and styrene, was recorded at 1100 °C. These gasses correspond to the relative toxicity of the resin for the environment and the human body in the conditions of foundry practice. The results of the chromatographic analysis of shungite are presented in Figure 9. At 250 °C outside CO_2_, no other gas emissions were recorded. At 1100 °C, three main pyrolysis gas products were registered, analogous to the products of thermal decomposition of coal dust, i.e., carbon dioxide, benzene, and toluene, among which CO_2_ was the most intensified. The intensity of benzene and toluene emissions is definitely lower compared to their emission during the pyrolysis of coal dust. Determining the quantitative discrepancy of gas pyrolysis products of analysed samples using the PY-GC/MS method, however, was not possible due to the construction of the device (no pyrolysis time control) and the size of the sample used for tests (several mg), which had an impact on the lack of Research repeatability during measurement [39]. On the Shungite chromatogram obtained at 1100 °C there was no presence of peaks of noticeable intensity near the CO_2_ peak, as was the case with a chromatogram of coal dust at the same temperature (Figure 8a). This shows that among the gasses products of the thermal degradation of shungite there are no significant quantities other than Benzene and toluene organic compounds. Shungite, therefore, remains the least emitting relationship in the considered temperature and group of coal additives.

### 3.4. Gasses Emission Tests

For gasses emission tests used a moulding sand with organobentonite characterized by the best strength and technological properties among moulding sands with SN/5PAA binder and participation of shungite (SZ), i.e., SN/5PAA/SZ/HCR. Considering the ecological aspect related to the possibility of replacing coal dust or hydrocarbon resin potentially the material with less emissions, such as shungite, SN/5PAA/P/HCR was used as a reference moulding sand. The gas emission of the new binding material and the carbon additive was compared with the level of gaseous emissions of the bentonite-Kormix blends practically used in the casting technology.

The course of changes in the volume of released gasses in time for all moulding sands was similar, but the lowest, and therefore the most advantageous in terms of technology, due to the limitation of the possibility of surface casting defects, was characteristic of the bentonite-Kormix blend. Table 6 shows the results of gasses emission measurements from the analysed moulding sand samples.

The gas volume generated from 1 kg of bentonite-Kormix moulding sand was 19.5 dm^3^, while for the newly developed moulding sans with the participation of organobentonite and P/HCR mixtures—27.3 dm^3^ and organobentonite with Sz/HCR—23.2 dm^3^/kg of moulding sand. The gas volume separated from the SN/5PAA/P/HCR moulding sand was therefore higher than that of the moulding sand in which the coal dust was replaced with a new additive (Sz), which is a direct consequence of the lower emissivity of gaseous products of thermal decomposition of shungite (due to its composition and properties), in relation to other lustrous carbon carriers, confirmed in the previous chapter. It should also be noted that the amounts of gasses generated during the thermal decomposition of carbon additives in all tested systems are at a similar level, so the use of the SN/5PAA/Sz/HCR system in synthetic moulding sands should not increase the propensity of the moulding sand to create casting defects. By analysing the gas evolution rate curves from the SN/5PAA/Sz/HCR and SN/5PAA/P/HCR moulding sands, it was found that the emission of gaseous products thermal decomposition of the components occurred the fastest in the first 50 s (Figure 10).

The visible maxima for the dependence of gas evolution rate as a function of time, recorded for all samples, were most likely related mainly to the release of aromatic hydrocarbons, the amount of which gradually decreased over time. The time of maximum gas evolution rate equal to 0.2 mL/g·s for SN/5PAA/Sz/HCR and 0.26 mL/g·s for SN/5PAA/P/HCR was 42 and 51 s, respectively. Moulding sand containing shungite, was characterized by the mildest course of gas evolution rate. It is advantageous from the technological point of view—a smaller volume of gasses flows in a given time unit in the intergrain spaces of the moulding sand. Thus, the likelihood of gasses being trapped in the solidifying casting is reduced due to the limited permeability of the compacted moulding sand.

### 3.5. Analysis of Gaseous Products (the Concentrations of BTEX Compounds)

Table 7 summarizes the results of concentrations of aromatic hydrocarbons compounds from the BTEX group (benzene, toluene, ethylbenzene, and o-, m-, p-xylenes) obtained for the considered moulding sands: SN/5PAA/Sz/HCR and SN/5PAA/P/HCR. The summary also includes the concentrations of these gaseous products generated from the bentonite-Kormix blend.

Based on the analysis of the experimental data obtained by gas chromatography (GC), it can be concluded that, regardless of the type of moulding sands, the largest quantitative share had benzene. The amount of the remaining compounds from the BTEX group was at a much lower level.

Benzene, among all hydrocarbons from the BTEX group, is characterized by the highest toxicity, including negative effects on human health and the environment, therefore the threshold limit value (TLV) indicated for it is the lowest in this group and amounts to 1.6 mg/m^3^. For the remaining BTEX, the maximum permissible concentrations for health in the work environment are as follows: TLV_toluene_ = 100 mg/m^3^, TLV_ethylbenzene_ = 200 mg/m^3^ i TLV_xylenes_ = 100 mg/m^3^.

Comparing the obtained results for the moulding sands in terms of benzene emission, it should be stated that the moulding sand with SN/5PAA/Sz/HCR was characterized by approximately 2 times lower C_6_H_6_ emission level compared to the moulding sand with the bentonite-Kormix blend. It is a positive and expected result for the foundry industry in terms of work safety and environmental protection.

The level of ethylbenzene determined during the thermal decomposition of the moulding sand components SN/5PAA/Sz/HCR was also about half lower compared to the amount generated from a bentonite-Kormix blend in the conditions of pouring the mould with liquid cast iron. Taking into account the environmental aspect of casting technology in synthetic moulding sands, the SN/5PAA/Sz/HCR may be an environmentally friendly alternative to the commonly used bentonite-carbon mixtures.

### 3.6. Test Casting

The research work was completed with an additional stage in the form of a test casting and preliminary assessment of its surface quality. The main purpose of introducing lustrous carbon carriers to the bentonite moulding sand is to prevent the mould from burning to the casting during pouring with liquid cast iron. Therefore, the key parameters determining the usefulness of the obtained organobentonite as a new binder material and shungite as a new carrier of the desired carbon structure are the surface roughness of the casting and dimensional accuracy. Limiting attention to these two casting quality indicators is related to the difficulty of precise interpretation of the phenomena occurring in a non-dried mould poured with liquid metal, which results directly from the high dynamics of physicochemical processes taking place on the metal-mould interface.

A comprehensive analysis of the influence of the tested materials on the possibility of the formation of external and internal defects of the casting is a very extensive topic, therefore the attention was focused only on the general assessment of the surface quality of castings made with the use of moulds made of moulding sand with SN/5PAA/Sz/HCR, i.e., organobentonite and a mixture of shungite/HCR resin (Figure 11c) and S/Kormix, i.e., bentonite-Kormix blend (Figure 11d).

A stepped model was used to assess the surface quality of castings made in selected moulding sands. Thanks to the variable wall thickness along the height of the model, it is possible to initially verify the effectiveness of carbon additives and the influence of the wall thickness of the casting on the quality of the casting surface.

## 4. Conclusions

Investigations of technological properties: permeability, compactability, Dietert flowability, and moulding friability with organobentonite SN/5PAA obtained in the first part of research showed comparable values of determinations as a function of moulding sand moisture content in relation to characteristics obtained for reference SN and model S green sands. At the same time, an unfavourable influence of the organic modification of bentonite on the strength properties of the moulding sands expressed as a decrease in green compressive and green tensile strength was noted, which indicates an unfavourable influence of the polymer wrapping around the bentonite grains inhibiting the swelling capacity of the mineral and limiting the binding properties of the system to the sorption capacity of the polymer only.

The introduction of shungite into the moulding sand system as a substitute for one of the two popular lustrous carbon carriers, i.e., coal dust or hydrocarbon resin, had a favourable effect especially on the strength properties of the green sands compared to the systems with a mixture of P and HCR additives, as well as the model moulding sand of practical application S/Kormix. Such a relationship was observed both for moulding sands bond with calcium bentonite and its organic modifier, although in the latter case, the positive effect of the new additive is more pronounced. The system with the best strength characteristics, i.e., SN/5PAA/Sz/HCR, most probably resulting from the highest compactability of the components, has been covered by a patent application—invention project entitled: “Bentonite-bonded moulding compound with carbon additive”, application no: P.439688. The presence of shungite in the systems under consideration did not significantly affect the technological properties of the moulding sands containing it.

Analysing the ecological aspect, the moulding sand with the addition of shungite is characterized by similar gas volume emission and lower emissions of toxic compounds from the BTEX group compared to the moulding sand with coal dust: SN/5PAA/P/HCR and a model bentonite-Kormix blend. This confirms the initial assumptions about the possibility of considering shungite in the context of an ecological replacement for popular coal additives in synthetic moulding sands. The use of the organobentonite and mix of shungite/HCR resin allows for a cast with much better accuracy of the model shape and very good quality surfaces. It should be noted that the castings were not subjected to any additional processing (raw casting after struggling). This is a beneficial signal primarily due to the limitation of the need for additional cleaning and mechanical processing of the casting. Despite the lack of data on the type of phenomena occurring at the interface of the analysed moulding sand with liquid metal, the results of macroscopic observation of the casting surface indicate the possibility of the tested binding material and carbon additives of sufficient coal form, capable of creating a permanent barrier for liquid metal during the casting process.

## Figures and Tables

**Figure 1 materials-16-01585-f001:**
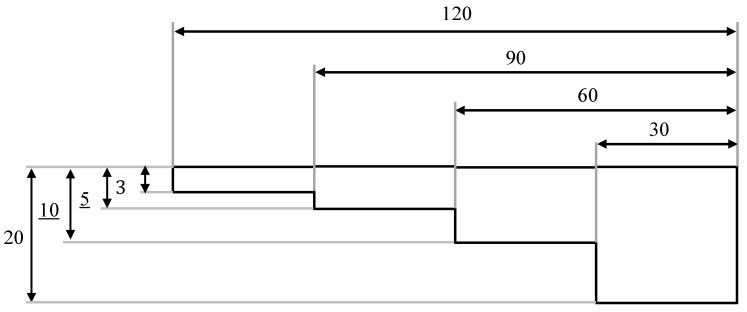
Scheme of the “stepped test casting” model (horizontal position).

**Figure 2 materials-16-01585-f002:**
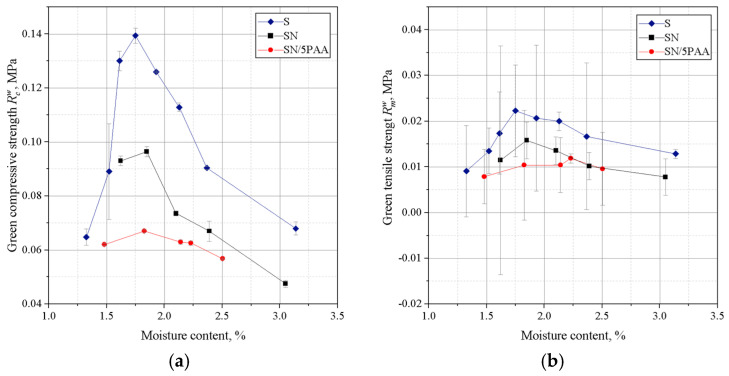
Green compressive (**a**) and green tensile strength (**b**) of moulding sands bonded with bentonites: S (sodium bentonite), SN (calcium bentonite), and organobentonite SN/5PAA.

**Figure 3 materials-16-01585-f003:**
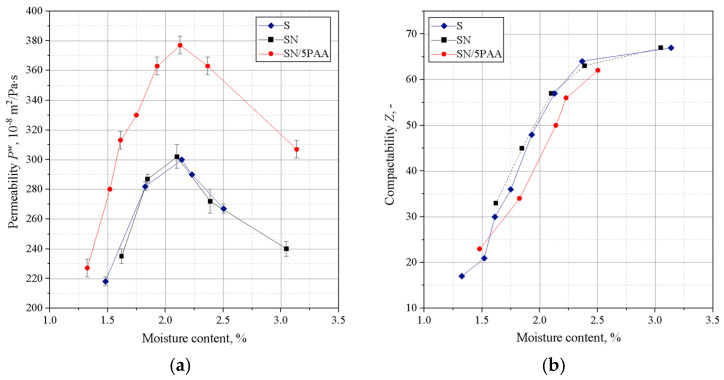
Permeability (**a**) and compactability (**b**) of green sands bond with bentonites S (sodium bentonite), SN (calcium bentonite), and organobentonite SN/5PAA.

**Figure 4 materials-16-01585-f004:**
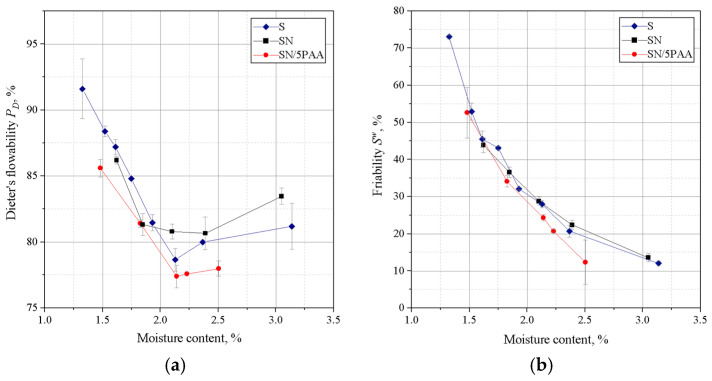
Dieter flowability (**a**) and friability (**b**) of green sands bond with bentonites S (sodium bentonite), SN (calcium bentonite), and organobentonite SN/5PAA.

**Figure 5 materials-16-01585-f005:**
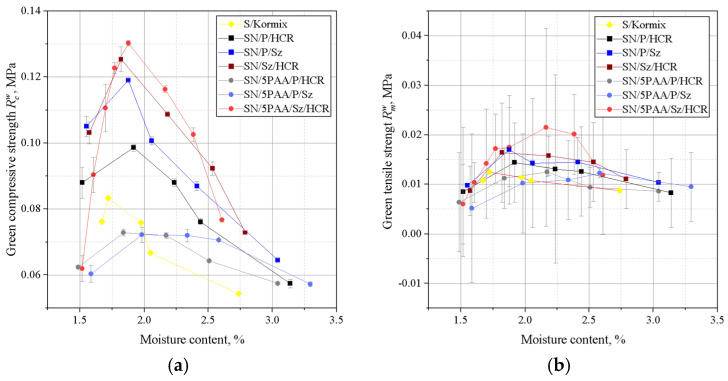
Green compressive (**a**) and green tensile strengths (**b**) of moulding sands bonded with calcium bentonite SN and organobentonite SN/5PAA with selected blends of carbon additives: coal dust (P), resin (HCR), and shungite (Sz). S/Kormix was used as a reference moulding sand (of practical use).

**Figure 6 materials-16-01585-f006:**
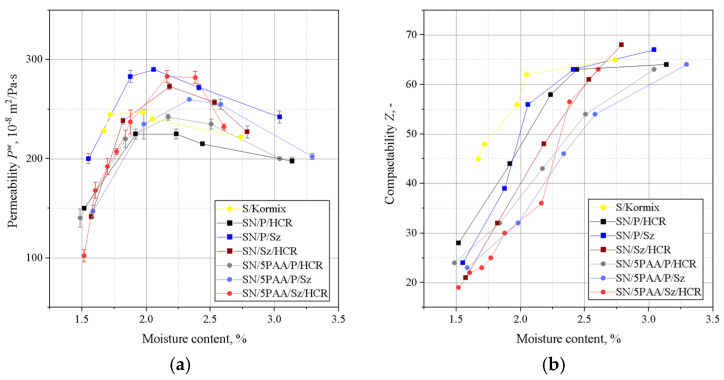
Green permeability (**a**) and green compactability (**b**) of moulding sands bond with calcium bentonite SN and organobentonite SN/5PAA with selected blends of carbon additives: coal dust (P), resin (HCR), and shungite (Sz). S/Kormix was used as a reference moulding sand (of practical use).

**Figure 7 materials-16-01585-f007:**
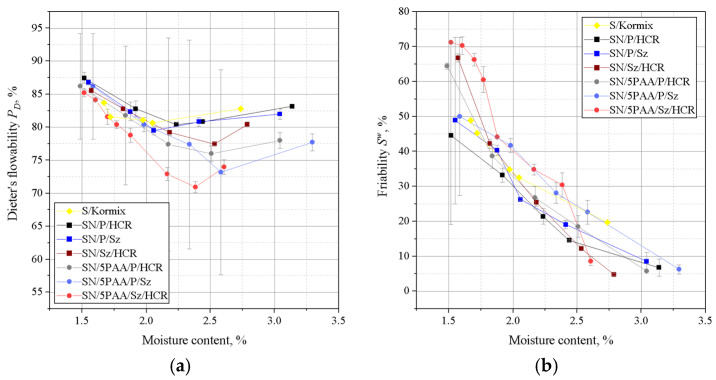
Dieter’s flowability (**a**) and friability (**b**) of moulding sands bond with calcium bentonite SN and organobentonite SN/5PAA with selected blends of coal additives: coal dust (P), resin (HCR), and shungite (Sz). S/Kormix was used as a reference moulding sands (of practical use).

**Figure 8 materials-16-01585-f008:**
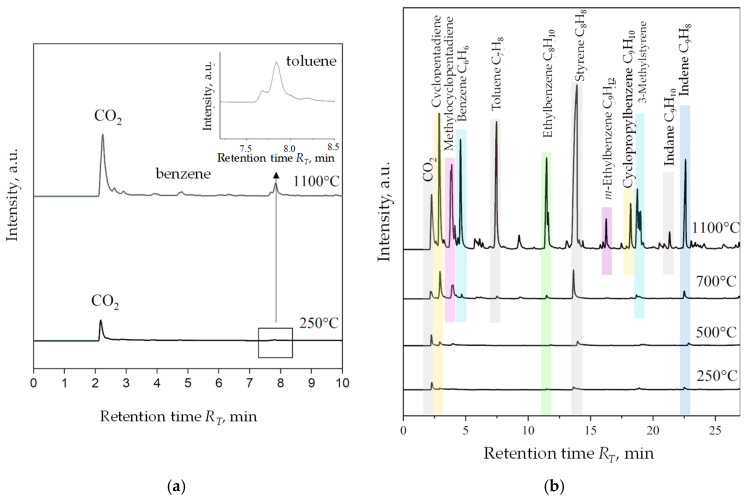
Chromatograms for coal dust (**a**), HCR resin (**b**).

**Figure 9 materials-16-01585-f009:**
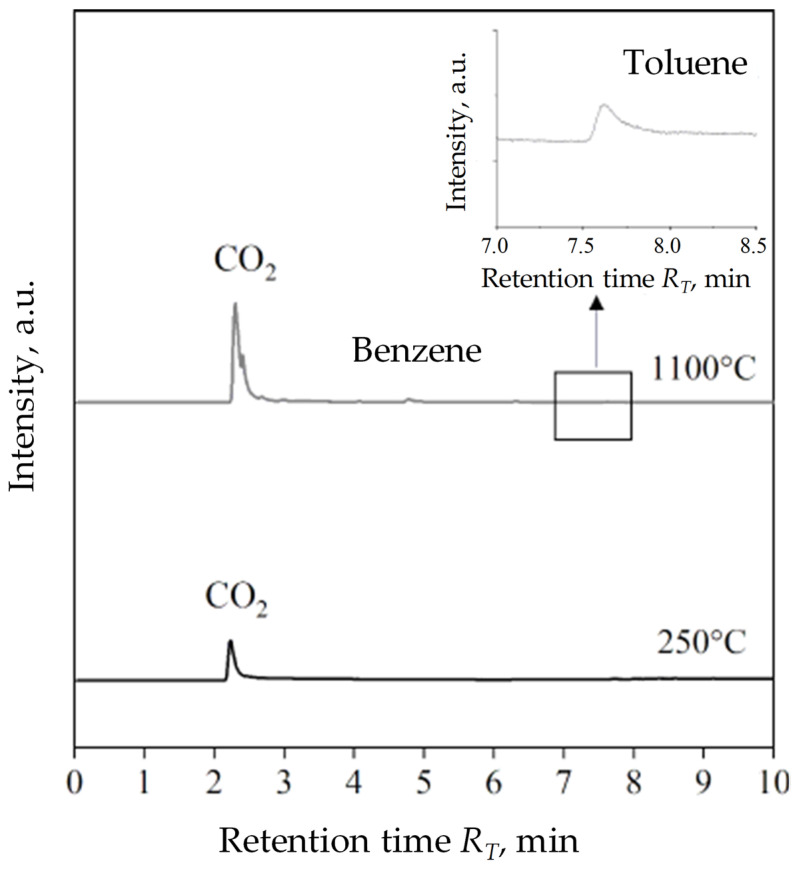
Chromatogram for shungite.

**Figure 10 materials-16-01585-f010:**
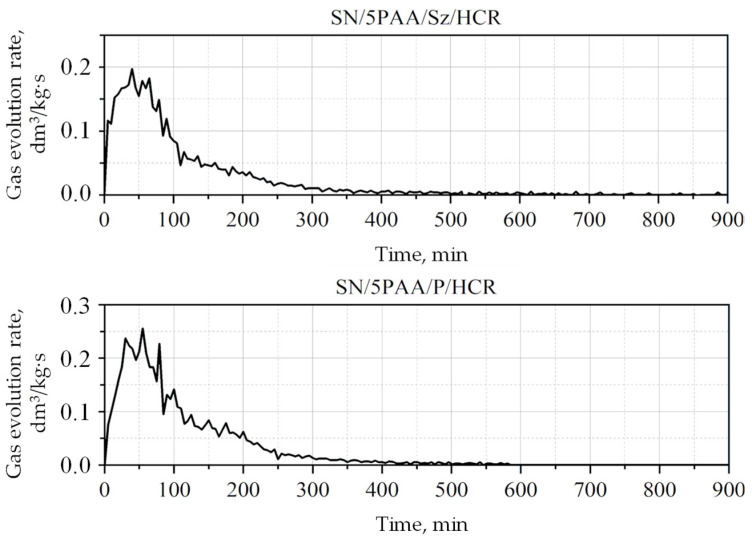
Gas evolution rate from tested moulding sands.

**Figure 11 materials-16-01585-f011:**
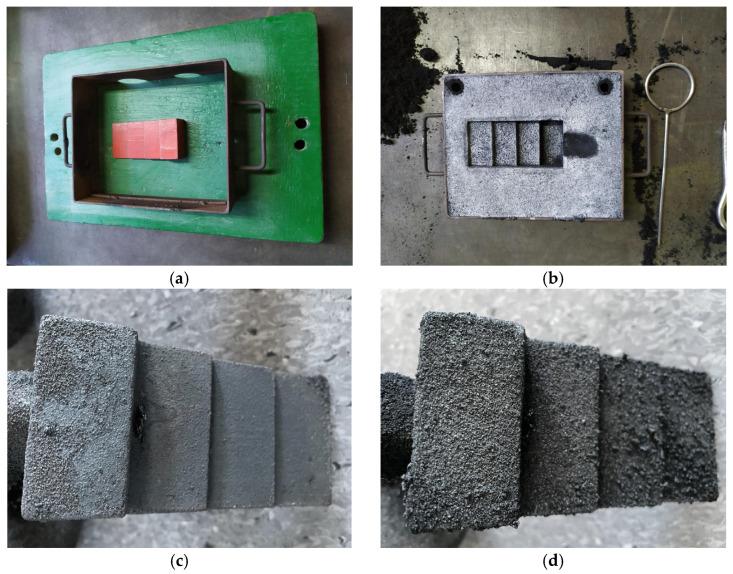
The process of making the casting: (**a**,**b**) preparing the mould, (**c**) macroscopic image of the iron casting made in a moulding sand with SN/5PAA/Sz/HCR without machining, (**d**) macroscopic image of the iron casting made in a moulding sand with S/Kormix blend without machining (right after breaking out of the mould).

**Table 1 materials-16-01585-t001:** Selected physicochemical properties of PAA according to the manufacturer’s safety data sheet.

Polymer/Structural Formula	Average Molar Mass *M_w_*, g/mol	Viscosity in 20 °C, cP	Form	pH
poly(acrylic acid) (PAA)	*M_w_* ~1800	≤2000	solid (powder)	2.4

**Table 2 materials-16-01585-t002:** Chemical composition and basic physicochemical parameters of calcium (SN) and sodium (S) bentonites [27].

	Chemical Composition		Humidity, %	pH (10% r-r)	Cation Exchange Capacity CEC, mmol/100 g	Montmorillonite (MMT) Content, %	Swelling Index *W_p_*, cm^3^/2 g	Carbonate Content, %
	Oxide Composition	Content, %						
SN	SiO_2_	67.39	8.1	9.0	65.3	69.2 (MMT content refers to active MMT)	8.7	1.5
	Al_2_O_3_	18.96						
	MgO	4.58						
	CaO	3.02						
	Fe_2_O_3_	2.73						
	Na_2_O	1.28						
	K_2_O	1.13						
	Sum	99.09						
S	SiO_2_	63.89	9.3	10.1	75.1	79.6 (MMT content refers to active MMT)	35.4	5.0
	Al_2_O_3_	18.50						
	MgO	3.54						
	CaO	3.06						
	Fe_2_O_3_	5.22						
	Na_2_O	3.37						
	K_2_O	1.32						
	Sum	98.90						

**Table 3 materials-16-01585-t003:** List of moulding sand ingredients.

Material	Role in the Moulding Sand	Supplier	Technical Data/Properties
Silica sand	sand grains (matrix)	Sibelco Poland	Main fraction 0.16–0.32 mm
Calcium bentonite (SN)	Binding material	ZGM “Zębiec” S.A (Starachowice, Poland)	Characteristics in Table 2.
Sodium bentonite (S)	Binding material	ZGM “Zębiec” S.A (Starachowice, Poland)	Characteristics in Table 2.
Organobentonite (SN/5PAA)	Binding material	Obtained in the course of research	
Hydrocarbon resin (HCR)	Carbon additive	ZGM “Zębiec” S.A (Starachowice, Poland)	carbon content (C): 98.5%; lustrous carbon content: min. 55%; moisture content: max. 0.4%; ash content: max. 0.5%; softening temperature: 95–115 °C; volatile fraction content: 95.4%
Coal dust (P)	Carbon additive	ZGM “Zębiec” S.A (Starachowice, Poland)	carbon content (C): 97.5%; content LC: min. 9%; moisture content: max. 0.5%; ash content: max. 5.6%; volatile fraction content: max. 36.0%
Shungite (Sh)	Carbon additive	Wessper	carbon content (C) based on XRF: 77.1%; other elemental composition based on XRF: 11.7% O; 8.9% Si; 0.7% Al.; 0.6% Fe; 0.4% K; 0.3% S; 0.1% Mg
Bentonite-Kormix blend (S/Kormix)	Binding material + Carbon additive	ZGM “Zębiec” S.A (Starachowice, Poland)	Manufacturer’s proprietary data

**Table 4 materials-16-01585-t004:** Composition of green sand with SN and S bentonites, SN/5PAA organobentonite, and carbon additives (in part by weight; pbw).

Green Sand—Labelling at Work	Sand Grains (Matrix), pbw	Binding Material, pbw	Coal Dust, pbw	HCR Resin, pbw	Shungite, pbw	Sum of Carbon Additives, pbw
SN	100	6	-	-	-	-
SN/P/HCR	100	6	2.2	1.4	-	3.6
SN/Sz/HCR	100	6	-	1.4	6.9	8.3
SN/P/Sz	100	6	2.2	-	4.0	6.2
SN/5PAA	100	6	-	-	-	-
SN/5PAA/P/HCR	100	6	2.2	1.4	-	3.6
SN/5PAA/Sz/HCR	100	6	-	1.4	6.9	8.3
SN/5PAA/P/Sz	100	6	2.2	-	4.0	6.2
S	100	6	-	-	-	-
S/Kormix	100	9.6 Part by weight of the bentonite-Kormix mixture				

**Table 5 materials-16-01585-t005:** Chemical composition of cast iron used in the experimental casting.

Chemical Composition	Fe	C	Si	Mn	P	S	Cr	Mg	Cu
%	92.998	3.71	2.69	0.44	0.05	0.01	0.04	0.042	0.02

**Table 6 materials-16-01585-t006:** Gasses emission from analysed moulding sands.

Moulding Sand	Gas Volume, dm^3^/kg Moulding Sand
SN/5PAA/Sz/HCR	23.2
SN/5PAA/P/HCR	27.3
Bentonite-Kormix blend (S/Kormix)	19.5

**Table 7 materials-16-01585-t007:** The concentrations of BTEX compounds.

Moulding Sand	Concentrations of BTEX Compounds, mg/kg Moulding Sand			
	Benzene	Toluene	Ethylbenzene	Xylenes
SN/5PAA/Sz/HCR	122.7	4.4	0.039	1.087
SN/5PAA/P/HCR	213.1	10.6	0.610	3.654
Bentonite-Kormix blend (S/Kormix)	254.7	9.6	0.062	0.390

## Data Availability

The data is contained within the article and/or available on request from the corresponding author.

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
