# Peer review of "Organobentonite Binder for Binding Sand Grains in Foundry Moulding Sands"

_materials, 2023, doi:10.3390/ma16041585_

Round 1

Reviewer 1 Report

The paper presents a very detailed experimental study. I especially appreciate the organized manner in which the manuscript is written, as well as the comprehensive and manuscript literature survey presented in the introduction section. The authors took into consideration a vast number of experimental factors in order to evaluate their materials.

Author Response

Dear Reviewer,

Thank you for the review. I am sending the answer to the review in the attachment.

With best regards,

Professor Beata Grabowska

Reviewer 2 Report

First, I must indicate that I am not an expert in foundry materials. I assume that from my fields of expertise, I have received this manuscript because the term “organobentonite” appears in the title and in the key words, and also montmorillonite in the key words. So, I have mainly checked these points in the manuscript.

However, when reading the manuscript, the information given on the organobentonite is very low. Some points are described in the Introduction, and the preparation of the organobentonite is described in lines 282-288, but the organobentonite is not characterized and its influence in the further materials and processes is not discussed. In fact, I do not find information that supports if the use of polyacrylic acid is actually positive for the final materials, if the amount considered is optimal, … So, from this point of view, claiming that the use of organobentonite binder is reported may be confusing for the readers.

Other points:

In lines 38-39 the same sentence is repeated twice. The first time, the formula “standard” for montmorillonite is given as Na+0,67 [Al3,33 Mg0,67]–(0,67) Si8O20(OH)4 · nH2O, but in the next line, the sentence is incomplete, and a parameter x is introduced but without giving its meaning.

In my opinion the introduction is too long.

The preparation of the organobentonite is confusing. In line 282 the authors talk of PAA solutions with percentage concentrations of 20%, 43%, 56%, at a ratio of 5 g of bentonite per 100 ml of distilled water. But three lines later they talk of 5%, 15% and 25% by weight of mineral. This must be clarified.

Author Response

(The authors gave the same response as above.)

Reviewer 3 Report

Lots of experiments and tests were performed, and some results have reference value. It can be accept after carefully modified the writting.

Such as, the introduction section needs further refinement, too much content, and the expression is not concise enough. I can not obtain the key point in this research area through the introduction.

Author Response

(The authors gave the same response as above.)

Reviewer 4 Report

Comments

11. The Introduction part is far too extensive.

22. Lines 34-37:  Bentonite is a rock, not a mineral, and therefore it has not a crystallographic structure.

33. There are some phrases that should be rephrased e.g. lies 38-41.

44. Materials and methods, Table 2: SN is not a bentonite but a bentonitic clay. The amount of montmorillonite is below 80% (69.2%). This is very significant in my opinion since the authors claim that they have worked with bentonite and their manuscript is based on that.

55. Materials and methods, Table 2: What are the other minerals present? From the chemical composition is not clear if the amounts of montmorillonite and carbonate are as the authors claim. The complete mineralogical composition should be provided as well as XRD patterns.

66. Table 5. I believe that Table 5 is incomplete. The amounts of some elements are missing (especially iron).

Author Response

(The authors gave the same response as above.)

Round 2

Reviewer 2 Report

The manuscript can now be accepted.